# Identification of new drugs to counteract anti-spike IgG-induced hyperinflammation in severe COVID-19

Chiara E Geyer[1],*, Hung-Jen Chen[1],*, Alexander P Bye[2,3,4], Xue D Manz[5], Denise Guerra[6], Tom G Caniels[6], Tom PL Bijl[6], Guillermo R Griffith[7], Willianne Hoepel[1], Steven W de Taeye[6], Jennifer Veth[1], Alexander PJ Vlaar[8], Amsterdam UMC COVID-19 Biobank[‡], Gestur Vidarsson[9,10], Harm Jan Bogaard[5], Jurjan Aman[5], Jonathan M Gibbins[2], Marit J van Gils[6], Menno PJ de Winther[7,†], Jeroen den Dunnen[1,†]

**Previously, we and others have shown that SARS-CoV-2 spike-specific IgG antibodies play a major role in disease severity in COVID-19 by triggering macrophage hyperactivation, disrupting endothelial barrier integrity, and inducing thrombus formation. This hyperinflammation is dependent on high levels of anti-spike IgG with aberrant Fc tail glycosylation, leading to Fcγ receptor hyperactivation. For development of immune-regulatory therapeutics, drug specificity is crucial to counteract excessive inflammation whereas simultaneously minimizing the inhibition of antiviral immunity. We here developed an in vitro activation assay to screen for small molecule drugs that specifically counteract antibody-induced pathology. We identified that anti-spike-induced inflammation is specifically blocked by small molecule inhibitors against SYK and PI3K. We identified SYK inhibitor entospletinib as the most promising candidate drug, which also counteracted anti-spike-induced endothelial dysfunction and thrombus formation. Moreover, entospletinib blocked inflammation by different SARS-CoV-2 variants of concern. Combined, these data identify entospletinib as a promising treatment for severe COVID-19.**

## Introduction

The ongoing severe acute respiratory syndrome coronavirus (SARS-CoV-2) pandemic is associated with millions of deaths and immense pressure on healthcare systems and economies worldwide ([1, 2]). In most patients, SARS-CoV-2 infection led to a mild manifestation of coronavirus disease 2019 (COVID-19) characterized by flu-like symptoms such as cough, fever, and fatigue. However, some patients, particularly in the unvaccinated population ([3]), develop severe and lethal complications including pneumonia, acute respiratory distress syndrome, thromboembolism, and sepsis ([4]). One characteristic of severe COVID-19 cases is the fast deterioration of the symptoms 1–2 wk after onset, accompanied by prolonged and elevated systemic pro-inflammatory cytokine levels, particularly interleukin (IL)-6, TNF, and IFNs ([2, 5, 6]). In addition to the hyperinflammatory states, severe COVID-19 patients develop multiorgan dysfunction that can be explained by derangements in hemostasis, also known as COVID-19-associated coagulopathy ([7, 8, 9]). Although the exact mechanisms of COVID-19–associated coagulopathy remain unclear, a complex interplay between coronaviruses, endothelial cells, platelets, elevated immune responses, and dysfunction of the coagulation system has been postulated ([10]).

Despite the increasing coverage of safe and effective vaccines worldwide, SARS-CoV-2 continues to spread rapidly. As the virus evolves, several variants of concern (VOC) characterized by increased transmissibility or virulence have been discovered ([11, 12, 13, 14]). Recent studies reveal a rapid increase in symptomatic COVID-19 cases in the vaccinated population, indicating reduced vaccine effectiveness over time and the emergence of new immune-escaping variants ([15, 16, 17]). Newly occurring virus variants to which previous vaccines do not provide sufficient protection are a threat to global public health ([18, 19]). Moreover, some people including immune-

[1]Center for Experimental and Molecular Medicine, Amsterdam Institute for Infection and Immunity, Amsterdam University Medical Centers, Amsterdam, Netherlands [2]Institute for Cardiovascular and Metabolic Research, and School of Biological Sciences, University of Reading, Reading, UK [3]Molecular and Clinical Sciences Research Institute, St George's University, London, UK [4]School of Pharmacy, University of Reading, Reading, UK [5]Pulmonary Medicine, Amsterdam University Medical Centers, Amsterdam, Netherlands [6]Medical Microbiology and Infection Prevention, Amsterdam Institute for Infection and Immunity, Amsterdam University Medical Centers, Amsterdam, Netherlands [7]Department of Medical Biochemistry, Amsterdam Cardiovascular Sciences, Atherosclerosis & Ischemic Syndromes, Amsterdam Institute for Infection and Immunity, Inflammatory Diseases, Amsterdam University Medical Centers, Amsterdam, Netherlands [8]Department of Intensive Care Medicine, Amsterdam Institute for Infection and Immunity, Amsterdam University Medical Centers, Amsterdam, Netherlands [9]Experimental Immunohematology, Sanquin Research, Amsterdam, Netherlands [10]Department of Biomolecular Mass Spectrometry and Proteomics, Utrecht Institute for Pharmaceutical Sciences and Bijvoet Center for Biomolecular Research, Utrecht University, Utrecht, Netherlands

Correspondence: j.dendunnen@amsterdamumc.nl; m.dewinther@amsterdamumc.nl
*Chiara E Geyer and Hung-Jen Chen contributed equally to this work and share first authorship
†Menno PJ de Winther and Jeroen den Dunnen contributed equally to this work and share last authorship
‡Amsterdam UMC COVID-19 Biobank members and affiliations of the Amsterdam UMC COVID-19 Biobank are listed in Table S1

compromised populations or patients receiving immunomodulatory medications develop poor vaccination responses ([20]).

Therefore, in addition to disease prevention by vaccination, efforts have been made to develop treatments to alleviate symptoms. Several effective anti-viral therapeutics are authorized for COVID-19 treatment. Molnupiravir, a prodrug of a ribonucleoside analog introducing replication errors ([21]), has been shown to hasten the elimination of infectious viruses ([22], [23]). Nirmatrelvir, a SARS-CoV-2 main protease inhibitor, together with the HIV-1 protease inhibitor ritonavir, has been developed as a combined treatment (Paxlovid), which largely reduces the risk of hospitalization or death ([24], [25]). Given that anti-viral treatments do not rectify the underlying excessive host immune response deteriorating COVID-19, studies have also focused on attenuating uncontrolled inflammation in severe cases. Dexamethasone is the first approved immunoregulatory therapeutic that significantly reduces the risk of death, particularly in patients requiring mechanical ventilation or supplemental oxygen ([26], [27]). The efficacy of steroids in treating critical COVID-19 cases supports the idea that immune components contribute to disease severity. However, although steroid therapy is a successful approach in suppressing excessive inflammation and dampening COVID-19 complications, concerns remain about secondary infection and the reactivation of latent infections ([28], [29], [30]). Furthermore, as a potent corticosteroid, dexamethasone has a significant impact on the immune system and could cause a delay in viral shedding and have consequences in various organs ([31], [32]). Therefore, there is still an unmet need for a specific immunomodulatory treatment that reduces uncontrolled inflammation while keeping the anti-viral response intact simultaneously.

Previously, we and others provided evidence that SARS-CoV-2 spike protein-specific immunoglobulin G (IgG) promotes excessive production of pro-inflammatory mediators by alveolar macrophages and monocytes, disrupts endothelial barrier function, and activates platelets, thereby contributing to the exacerbation of COVID-19 in severe cases ([33], [34], [35]). The pathogenic effect mediated by anti-spike IgG is induced via the overactivation of fragment crystallizable region γ receptors (FcγRs) on innate immune cells ([5], [33], [36]). Two specific antibody features of severe COVID-19 patients contribute to the excessive immune response: extremely high anti-spike IgG titers and aberrant glycosylation of the IgG Fc tail, which when combined lead to the overactivation of FcγRs. The over-activated macrophages create a pro-inflammatory environment that leads to endothelial dysfunction and platelet adhesion. Furthermore, the aberrantly glycosylated IgG together with spike protein can form immune complexes that directly enhance platelet thrombus formation ([37]).

Spleen-associated tyrosine kinase (SYK) is a critical component in FcγR signal transduction ([38]) and hence serves as a potential target. The SYK inhibitor R406 (the active form of FDA- and EMA-approved drug fostamatinib) has been recently identified as an effective immunoregulatory drug modulating the activities of immune cells and platelets in severe COVID-19 ([33], [37], [39], [40]) and has been applied in several clinical trials (NCT04581954, NCT04629703, NCT04924660) ([41]). Once SYK is activated, it binds to phosphoinositide 3-kinase (PI3K) and triggers downstream signaling cascades ([42], [43]). Although the SYK–PI3K axis drives macrophage chemotaxis and phagocytosis ([38], [44], [45]), ample evidence

shows that SYK–PI3K activation also promotes the expression of inflammatory mediators ([46], [47], [48], [49]). Furthermore, the SYK–PI3K signaling pathway also contributes to platelet activation, adhesion, and aggregation ([50]). Therefore, interventions targeting SYK and PI3K activity might provide potential treatment options for severe COVID-19.

In this study, we set out to identify inhibitors counteracting immune complex-induced hyperinflammation. We developed a macrophage activation assay capable of determining compound potency and efficacy against anti-spike-specific inflammation. We applied this screening assay on approved and investigational small molecule inhibitors. We demonstrate that several SYK and PI3K inhibitors can counteract the hyperinflammatory state induced by anti-spike immune complexes. We identify entospletinib, a SYK inhibitor, as a promising candidate drug to tackle anti-spike IgG-mediated inflammation, endothelial barrier disruption, platelet adhesion, and thrombus formation. Moreover, entospletinib dampens the anti-spike IgG-mediated inflammation induced by different VOCs.

## Results

### Anti-spike IgG-induced inflammation can be specifically counteracted by targeting SYK

To quantify the potency and selectivity against anti-spike-mediated inflammation, we determined the half-maximal inhibitory concentration ($IC_{50}$) on macrophage activation. Previously, our transcriptomic classification showed that M-CSF and IL-10-differentiated macrophages most closely resemble human primary alveolar macrophages ([51]). We applied these monocyte-derived alveolar macrophage-like macrophages (MDAMs) in the assay. Briefly, MDAMs were treated with different compounds at increasing concentrations 30 min before stimulation by the TLR3 ligand polyinosinic:polycytidylic acid (poly(I:C)) (a viral stimulus mimic) in the presence or absence of recombinant anti-spike IgG-formed immune complexes (Fig 1A). We assessed the pro-inflammatory activity of macrophages by measuring IL-6 production. We hypothesized that if the compound is specific for FcγR signaling, it will dose-dependently decrease anti-spike-dependent IL-6 production, whereas leaving the activation by poly(I:C) alone unchanged. We investigated two SYK inhibitors R406 (the active form of fostamatinib) and entospletinib, along with the standard-of-care drug dexamethasone. Dose-dependent inhibitory curves were then plotted and the $IC_{50}$ values were calculated for each inhibitor for the two stimulation conditions (Fig 1B–D).

All compounds suppressed IL-6 production by macrophages upon co-stimulation by poly(I:C) and anti-spike immune complex (red curves in Fig 1B–D). Dexamethasone showed the best potency with the lowest concentration (around 20–100 nM) required to achieve maximal inhibition, compared with 0.5–1 $\mu$M for R406 and entospletinib. Notably, dexamethasone similarly blocked anti-spike-induced and virus-induced IL-6 production (average $IC_{50}$ = 3.6 or 4.4 nM with or without anti-spike IgG, respectively) (Fig 1B). Compared with dexamethasone, both SYK inhibitors exerted greater

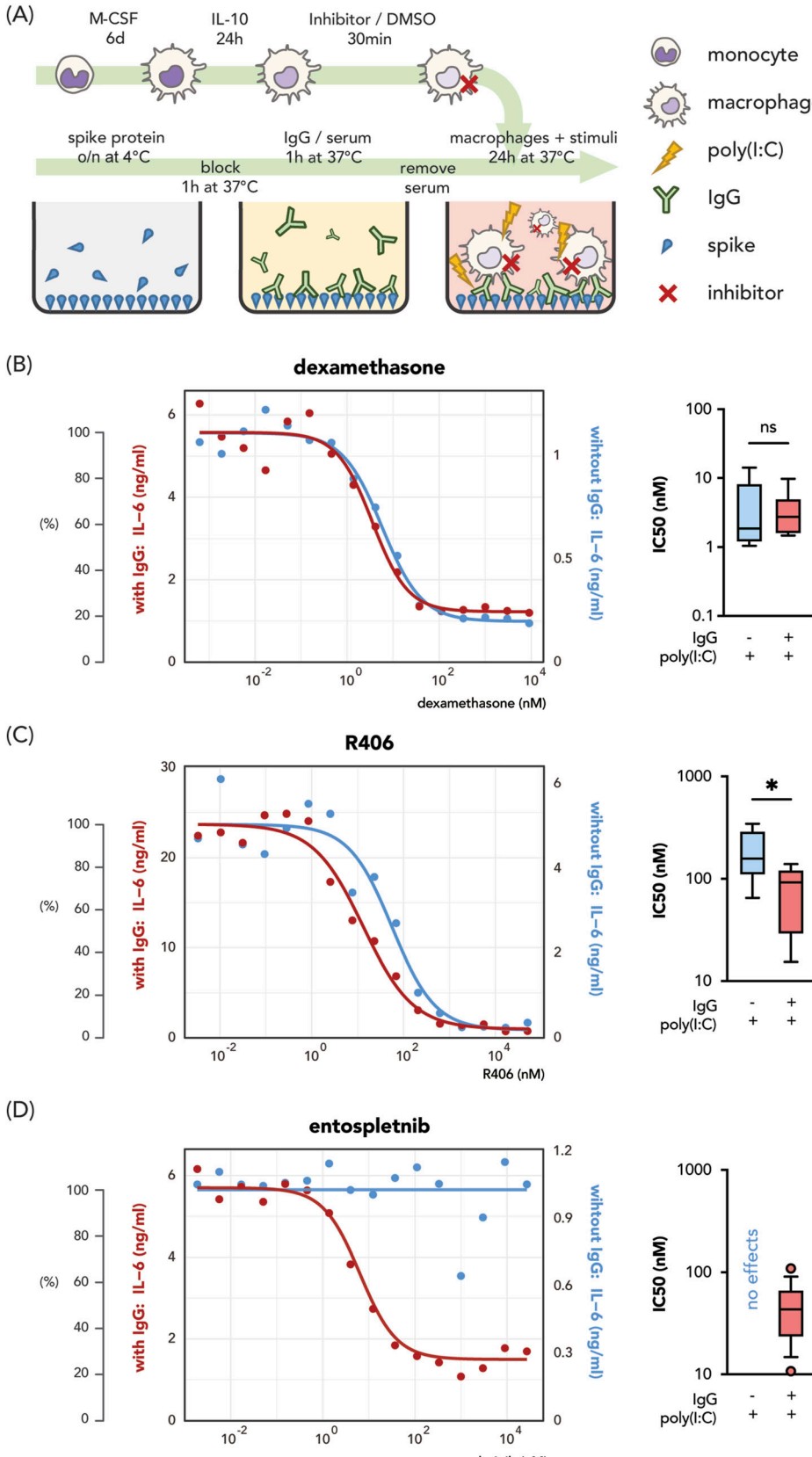

**Figure 1. Immunoregulatory activities of dexamethasone and SYK inhibitors R406 and entospletinib on IL-6 production by stimulated macrophages.**
**(A)** Schematic overview of the experimental setup. Monocyte-derived alveolar macrophage-like macrophages were generated by differentiating peripheral monocytes with M-CSF and IL-10. The generated Monocyte-derived alveolar macrophage-like macrophages were then treated with inhibitors in increasing concentration or DMSO 30 min before stimulation with viral stimulus poly(I:C) with or without the presence of immune complexes. Immune complex is formed by plate-bounded SARS-CoV-2 spike proteins and monoclonal anti-spike IgGs. All conditions are with SARS-CoV-2 spike proteins. **(B, C, D)** IL-6 production was used as the pro-inflammatory activation readout. **(B, C, D)** Representative data of macrophage activation assay for (B) dexamethasone, (C) R406, and (D) entospletinib, with the left Y axis and red curves showing the concentration measured from poly(I:C) and anti-spike immune complex conditions and right Y axis and blue curves activation with poly(I:C) alone. Half-maximal inhibitory concentrations (IC$_{50}$) from different macrophage donors (dexamethasone [n = 6], R406 [n = 5], and entospletinib [n = 14]) per stimulation condition are plotted as box plots indicating 10–90 percentile and median. Significant differences were calculated with a paired $t$ test. *$P < 0.05$.

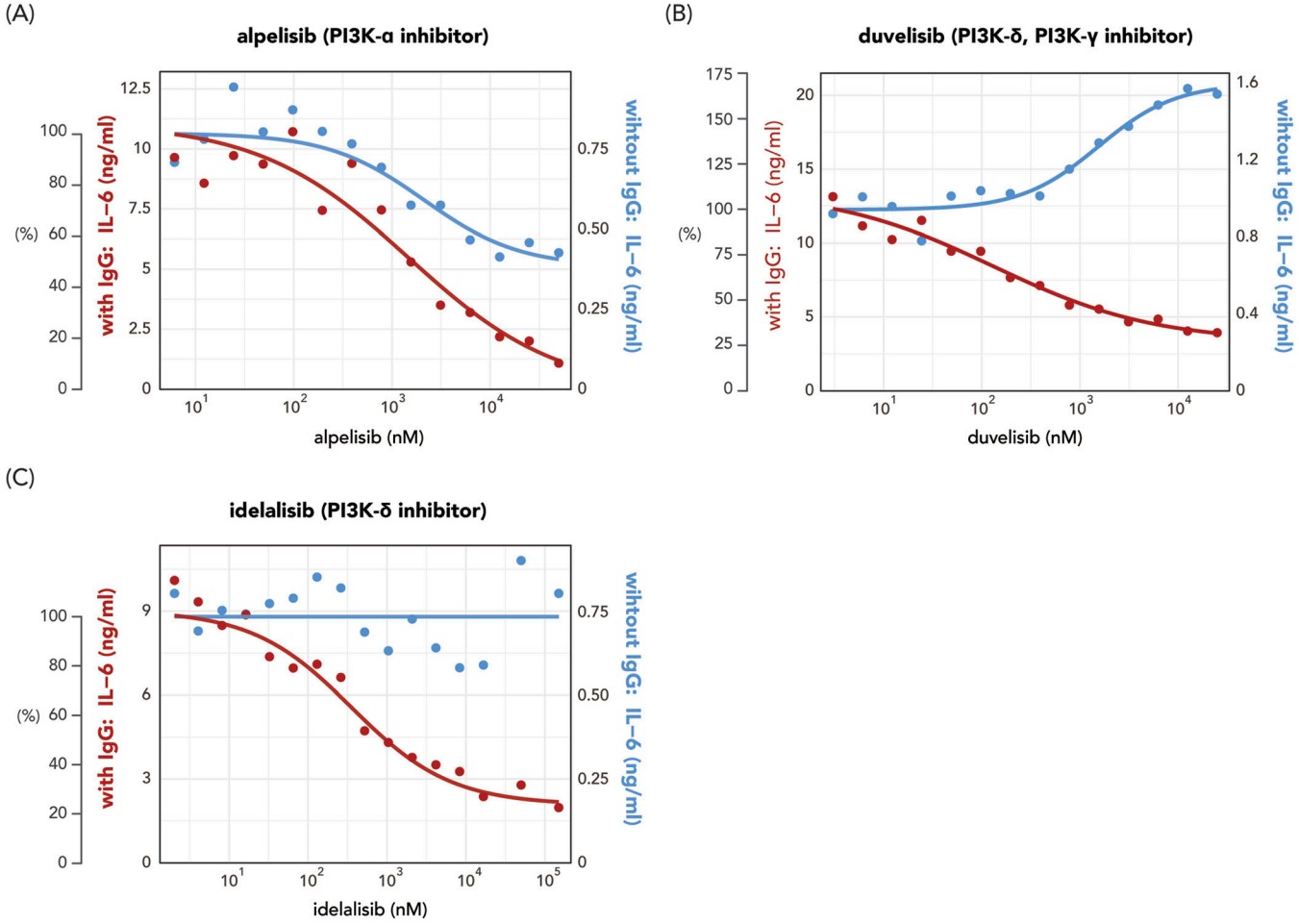

**Figure 2. Immunoregulatory activities of PI3K inhibitors on IL-6 production by stimulated macrophages.**
**(A, B, C)** Representative data of macrophage activation assay for (A) the PI3K$\alpha$ inhibitor alpelisib, (B) the PI3K$\delta$ and PI3K$\gamma$ inhibitor duvelisib, and (C) the PI3K$\delta$ inhibitor idelalisib.

potency for anti-spike-mediated inflammation. We observed a significant difference between $IC_{50}$ values for the R406 treatment against anti-viral and anti-IgG-induced IL6 production (mean $IC_{50}$ value of 191.9 nM for poly(I:C) alone-induced IL-6 and 78.5 nM for anti-IgG and poly(I:C) co-stimulation) (Fig 1C). Entospletinib was the most anti-spike-dependent inflammation-specific compound which did not affect poly(I:C)-only activated macrophages, and exhibited higher potency than R406 ($IC_{50}$ = 45.6 nM, Fig 1D).

### PI3K inhibitors affect macrophage activation

Next, we investigated the effect of inhibitors targeting PI3K, a downstream kinase in the Fc$\gamma$R-SYK signaling pathways. We carried out the same macrophage activation assay used for SYK inhibitors with compounds inhibiting different PI3K isoforms. In general, compared with SYK inhibitors, PI3K inhibitors required higher concentrations (>10 $\mu$M) to reach an 80% inhibition of anti-spike-induced IL-6 (Fig 2A–C). The effect on IL-6 induced by poly(I:C) alone varied between different compounds. Alpelisib, a PI3K-$\alpha$ inhibitor, inhibited IL-6 production with higher potency against anti-spike-

dependent inflammation in comparison to other tested PI3K inhibitors (Fig 2A). Interestingly, although PI3K-$\gamma$/$\delta$ inhibitor duvelisib suppressed macrophage IL-6 production in response to poly(I:C) and anti-spike immune complex co-stimulation, it amplified IL-6 secretion dose-dependently when only poly(I:C) was applied (Fig 3B). This observation suggests distinct regulatory functions for different PI3K isoforms in inflammatory processes and/or potential off-target effects of the drug. Another PI3K-$\delta$ inhibitor idelalisib counteracted anti-spike-dependent IL-6 production while not affecting the anti-viral response (Fig 2C). However, with the highest two concentrations tested in our assay, we observed reduced viability (data not shown), and an increase in IL-6 levels in the poly(I:C)-only condition. These results indicate that the potency of PI3K inhibitors is inferior to SYK inhibitors.

### Entospletinib counteracts serum-induced hyperinflammatory response by alveolar macrophages

We next assessed the effects of all tested inhibitors with their maximal inhibition concentrations against anti-spike-induced IL-6.

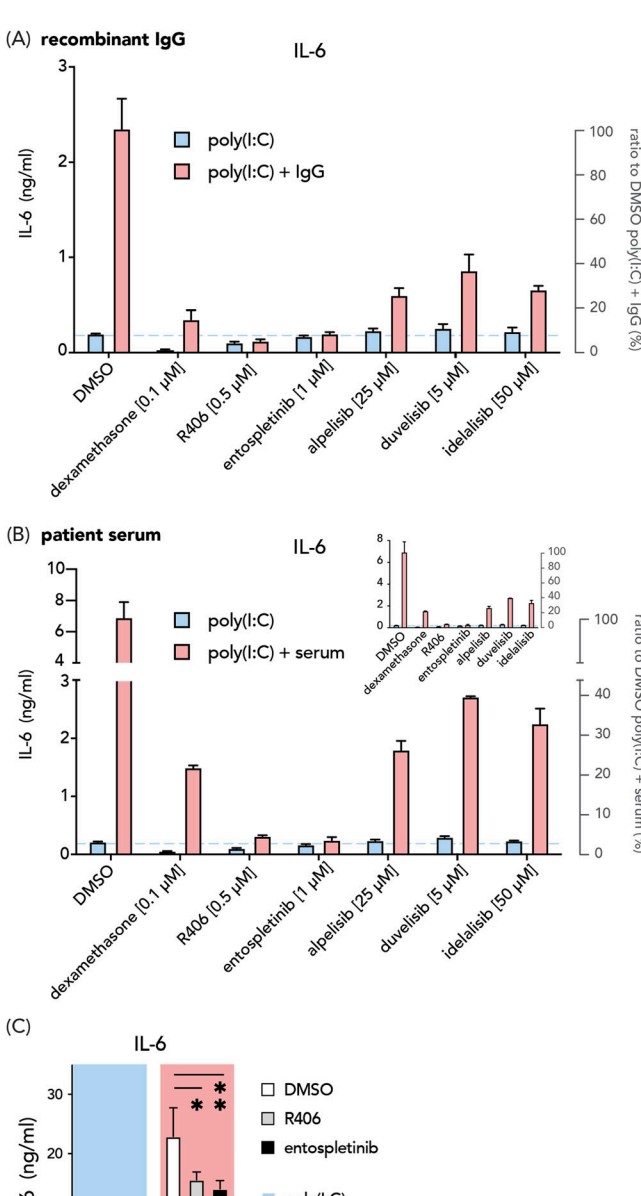

**Figure 3. Entospletinib counteracts serum-induced hyperinflammatory response by alveolar macrophages.**
**(A, B)** Representative data from four independent experiments showing IL-6 production by macrophages treated with dexamethasone and different SYK or PI3K inhibitors upon poly(I:C) stimulation with (red bars) or without (blue bars) immune complexes derived from a monoclonal antibody (A) or patient serum (B). Bar charts with one-segment Y axis (insert) or enlarged two-segment Y axis. **(C)** IL-6 production in DMSO, R406 or entospletinib-treated ex vivo bronchoalveolar lavage fluid-derived alveolar macrophages. Statistics were calculated using a two-way ANOVA and corrected using Tukey's multiple comparison test. *$P < 0.05$; **$P < 0.01$. n = 3 technical replicates per group, one representative example of n = 3 bronchoalveolar lavage donors. Data are shown as mean + SD.

In concordance with the dose-dependent assays, all treatments resulted in a substantial reduction in IL-6 production by macrophages upon anti-spike and poly(I:C) co-stimulation (red bars in Fig 3A). Dexamethasone and SYK inhibitors showed better potency with more profound effects at the selected concentration than PI3K inhibitors for blocking anti-spike-induced macrophage activation. More importantly, whereas dexamethasone hampered both anti-spike and anti-viral responses, SYK and PI3K inhibitors had limited impact on the IL-6 production in the poly(I:C)-alone condition (blue bars in Fig 3A). These results indicate that compounds deactivating SYK and PI3K serve as more selective treatment options for counterbalancing excessive inflammation induced by anti-spike immune complexes.

Unlike recombinant monoclonal antibodies, anti-spike IgGs in the patient serum are a pool of polyclonal antibodies against different domains of the spike protein with variate affinities and posttranslational modifications. Therefore, the immune complexes formed by recombinant monoclonal antibodies and serum could exert different biological activities. To assess whether SYK and PI3K inhibitors can counteract macrophage hyperactivation by serum-derived immune complexes, we generated spike–IgG immune complexes by incubating the spike protein with sera obtained from severely ill COVID-19 patients hospitalized at Amsterdam UMC from the first wave in early 2020. These patients were infected with the Wuhan strain and without prior vaccination. The sera were collected at the time of admission to the ICU. We observed similar inhibition patterns for all compounds compared with their monoclonal IgG counterparts (Fig 3B). SYK inhibitors R406 and entospletinib completely blocked anti-spike-induced IL-6 production, which dampened the cytokine levels to the concentration of the poly(I:C) condition (blue dashed line in Fig 3B). Interestingly, dexamethasone appeared to be less potent in blocking IL-6 induced by serum-derived anti-spike immune complexes than the ones formed by monoclonal IgGs (Fig 3A and B).

To ensure that the described decrease in pro-inflammatory cytokine induction is not influenced by off-target effects on cell viability, we measured the influence of the selected SYK and PI3K inhibitors on membrane integrity of alveolar macrophages. For all tested SYK and PI3K inhibitors, no significant effect on cell viability was observed in the tested concentration (Fig S1A and B).

Finally, we validated our findings in an ex vivo setting for the two most promising candidate compounds, by activating human alveolar macrophages obtained from bronchoalveolar lavage. Upon serum-derived immune complex activation, both R406 and entospletinib yielded comparable inhibition in bronchoalveolar lavage macrophages as the in vitro models (Fig 3C). To conclude, these data indicate that blocking SYK signaling can serve as a potent strategy against hyperactivation of alveolar macrophages induced by serum-derived immune complexes.

## Entospletinib dampens anti-spike IgG-associated pulmonary endothelial barrier disruption and thrombus formation

Pulmonary endothelial damage in COVID-19 is associated with macrophage activation and accumulation in the lungs (52).

Overactivated alveolar macrophages create a pro-inflammatory mi-lieu that subsequently promotes microvascular thrombosis and en-dothelial barrier disruption (53, 54, 55). We hypothesized that disrupted pulmonary endothelial function could be rescued by dampening macrophage hyperinflammatory activities with entospletinib. To in-vestigate this, we treated human pulmonary microvascular endothelial cells (HPMVECs) with conditioned media from activated MDAMs. We monitored the trans-endothelial electrical resistance of the HPMVECs monolayer over time as a readout of endothelial integrity.

In line with our previous findings in pulmonary artery endo-thelial cells (33), a prolonged disruption of endothelial barrier integrity was observed in HPMVECs treated with the conditioned media from macrophages co-stimulated with immune complexes generated with serum from severe COVID-19 patients and the viral stimulus (i.e., poly (I:C)) (the red thin line in Fig 4A). The conditioned media from poly(I:C)-only activated macrophages exerted a tran-sient effect on endothelial barrier function (the thin blue line in Fig 4A). Entospletinib was able to block anti-spike–mediated long-term

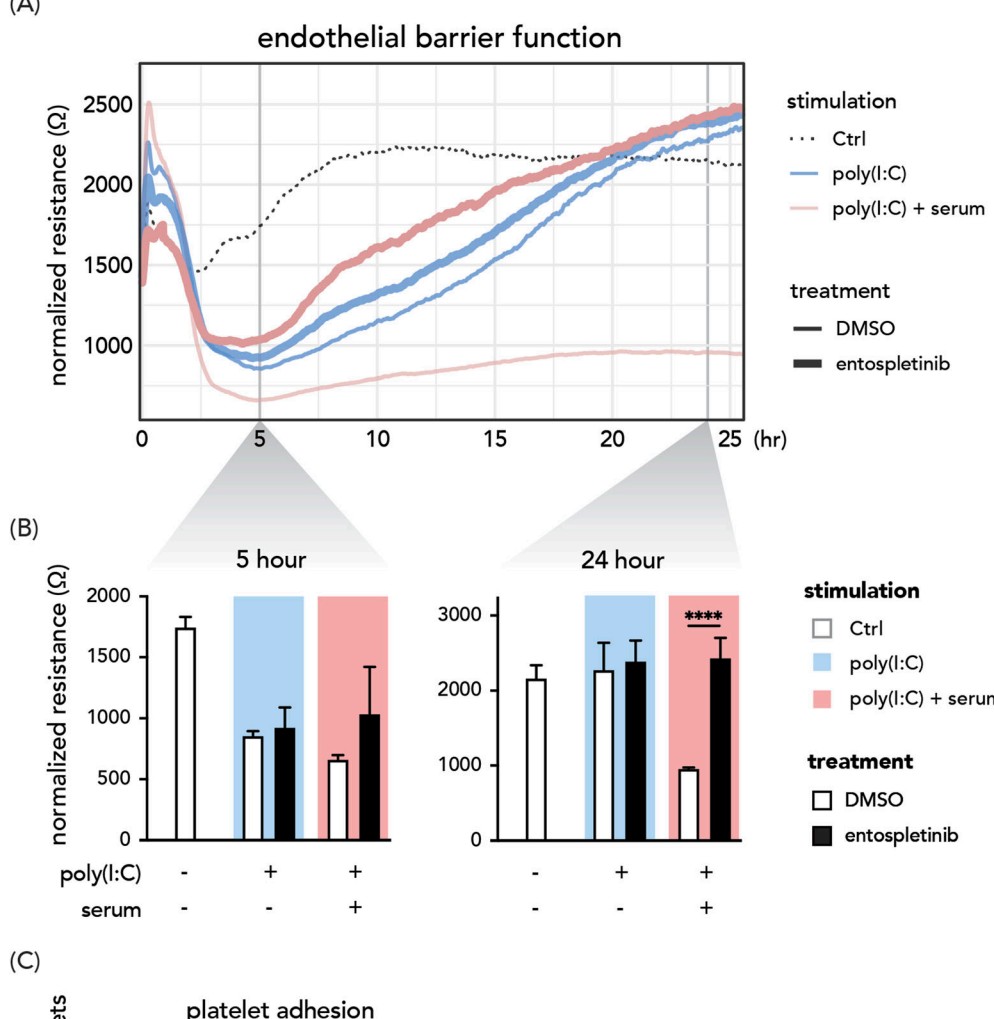

(A)

**Figure 4. Entospletinib dampens anti-spike IgG-associated pulmonary endothelial dysfunction and thrombus formation.**
**(A, B)** Representative data of trans-endothelial electrical resistance of the HPMVEC monolayer from two donors over time. HPMVECs were stimulated with conditioned media from activated macrophages treated with entospletinib or DMSO. The conditioned medium from macrophages without poly(I:C) or serum activation was used as a negative control. **(C)** Stimulated HPMVECs were perfused with platelets for 5 min, after which the area covered by platelets was quantified. n = 3 donors per group. Background colors in the bar plots indicate the stimulation the macrophages received. White or black bars indicate the drug treatments. Data are shown as mean + SD. Statistical significance was calculated using a two-way ANOVA and corrected using Tukey's multiple comparison test. ***$P$ < 0.001; ****$P$ < 0.0001.

endothelial dysfunction and significantly restore endothelial barrier integrity (thick red line in Fig 4A and B). Notably, entospletinib treatment did not affect HPMVECs stimulated with a supernatant of macrophages activated only by viral stimulus (the blue lines, Fig 4A). This indicates that entospletinib can selectively counteract the barrier-damaging mediators produced by macrophages upon stimulation with viral stimulus and serum-derived anti-spike immune complexes.

Next, we accessed the in situ thrombus formation by adding thrombocytes to macrophage-conditioned medium–activated HPMVECs under flow conditions (flow shear rate 2.5 dyn/cm$^2$). During perfusion, platelets adhered less to the HPMVECs exposed to conditioned media of entospletinib-treated macrophages under poly(I:C) and serum co-activation (Fig 4C). To sum up, we show that blocking FcγR signaling with entospletinib reduces pulmonary endothelial dysfunction and microvascular thrombosis formation.

### Entospletinib reduces aberrantly fucosylated ant-spike IgG-induced platelet activation

Recent evidence shows that anti-spike IgG of severely ill COVID-19 patients do not only indirectly activate blood platelets (via macrophages and endothelial cells), but also directly enhance platelet activation and thrombus formation (37). This direct activation of platelets critically depends on the aberrant IgG Fc tail glycosylation pattern that is observed in severely ill COVID-19 patients (34, 35, 56, 57). Whereas immune complexes with normal glycosylation patterns do not affect platelet adhesion, aberrantly glycosylated IgG–spike immune complexes enhance platelet activation in the presence of von Willibrand factor (vWF). As platelet activation by IgG is induced via FcγRIIa and the rapid phosphorylation of SYK (58), we studied the direct effect of entospletinib on platelets. We examined platelet adhesion under flow on coverslips coated with vWF and spike–IgG immune complexes formed by recombinant monoclonal anti-spike IgG COVA1-18 bearing aberrant glycosylation (9.1% fucosylated and 77.6% galactosylated). Platelets were pre-treated with entospletinib or DMSO before perfusion. Slides coated with vWF and spike and wild-type COVA1-18 immune complexes (97.8% fucosylated, 19.6% galactosylated) were used as a control (37). By quantifying the volume of thrombi, we show that aberrantly glycosylated immune complexes synergized platelet adhesion to vWF. Entospletinib counteracted the enhanced thrombus formation and reduced thrombus volume to the level of wild-type COVA1-18 controls (Fig 5A and B). These data demonstrate that entospletinib can reduce microvascular thrombosis induced by pathogenic platelet activation mediated by aberrantly glycosylated immune complexes.

### Antibody-induced inflammation is a shared mechanism across SARS-CoV-2 VOC and can be counteracted by SYK inhibitors

SARS-CoV-2 evolves to evade antibodies with mutations of the spike proteins (59). First, we investigated whether spike–IgG immune complexes of different SARS-CoV-2 VOCs induce hyper-inflammation by alveolar macrophages. We generated spike proteins of α, β, γ, δ VOCs, and the original Wuhan strain (GenBank accession MN908947.3) (60, 61). These spike proteins were subsequently applied

to form variant-specific immune complexes with COVA1-16, a monoclonal antibody that binds a highly conserved epitope on the spike receptor-binding domain (62). Immune complexes of all tested VOCs in the combination of poly(I:C) led to increased IL-6 release (Fig 6A) by macrophages. Next, we examined the effects of SYK inhibitors in counteracting anti-spike-dependent inflammation. SYK inhibitors R406 and entospletinib effectively suppressed the IL-6 production induced by immune complexes by 75–95 percent against all tested VOCs (Fig 6B). These data indicate that anti-spike-induced hyper-inflammation is a shared mechanism across different SARS-CoV-2 VOCs, which can all be blocked by SYK inhibition.

## Discussion

There is still an unmet need for specific, cost-effective, and orally bioavailable therapeutics to prevent disease progression to severe COVID-19. Here, we identify the small-molecular SYK inhibitor entospletinib as a potential medication with high potency and efficacy in specifically diminishing uncontrolled macrophage inflammation induced by anti-spike IgG immune complexes. Anti-spike IgG immune complexes can trigger the production of pro-inflammatory mediators, such as IL-6, TNF, and IFNs by alveolar macrophages (33). The high level of IL-6 produced by macrophages is a hallmark of COVID-19 (63). It has been shown that IL-6 induces oxidative stress, endothelial dysfunction, and coagulation cascade activation (64, 65). IL-6 receptor blockade treatments have been recommended by the WHO to tackle systemic inflammation in severe COVID-19 (66, 67). Given the critical role of SYK in FcγR signaling, blocking SYK activity could serve as a potential therapeutic for severe COVID-19 by ceasing the pathogenic hyper-activation of immune cells and the ensuing endotheliopathy (68).

The small molecule drug fostamatinib (the prodrug form of R406) is currently indicated for chronic immune thrombocytopenia because of its ability to block SYK signaling thus preventing the phagocytosis-based, antibody-mediated platelet destruction (69). Whereas mild thrombocytopenia is a common clinical manifestation in COVID-19 patients (70), immune thrombocytopenia can occur secondary to COVID-19 in both acute and late stages, particularly in old and severely ill patients (71). Therefore, fostamatinib might provide additional benefits apart from its immunosuppressive effect against anti-spike-specific inflammation. In severe or critical COVID-19 cases, clinical improvements were observed in the fostamatinib treatment group in a phase-II randomized trial (NCT04579393) (41). Based on this success, fostamatinib is currently tested in several phase-III clinical trials. However, the adverse effects of fostamatinib have been reported in cancers and rheumatoid arthritis and are attributed to off-target effects (69, 72, 73). Therefore, a more selective SYK inhibitor could provide better tolerability.

Entospletinib is a highly selective and orally efficacious second-generation SYK inhibitor (74). Although both tested SYK inhibitors can dampen anti-spike-induced inflammation, compared with R406, our data indicate that entospletinib has less effect on the macrophage anti-viral response, thereby representing a promising therapeutic approach for COVID-19 treatment. Notably, the average IC$_{50}$ value of entospletinib against anti-

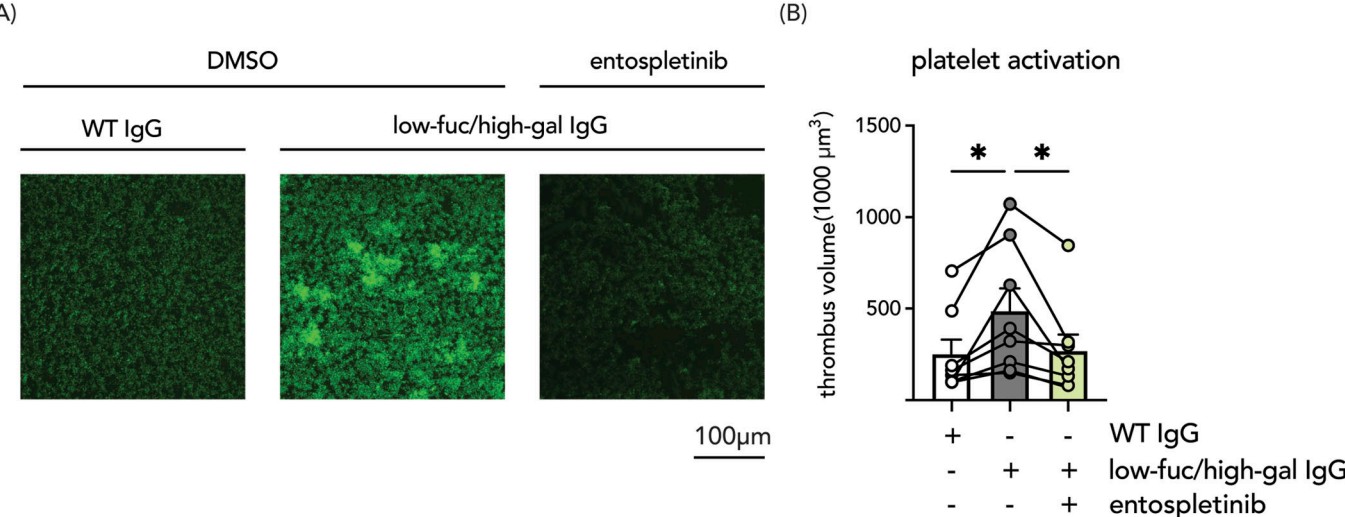

**Figure 5.  Entospletinib reduces aberrantly fucosylated ant-spike IgG-induced platelet activation.**
Thrombi formed under flow on vWF and spike–IgG immune complex-coated slides in perfusion chambers. Immune complexes were formed with normally glycosylated (WT) or lowly fucosylated and highly galactosylated (low-fuc/high-gal) IgGs. Platelets were pre-treated with either vehicle control (DMSO) or entospletinib (1 $\mu$M). **(A)** Representative images of thrombi stained with DiOC$_6$ (acquired at ×20 original magnification). **(B)** Quantification of thrombus volume from eight different platelet donors. Data are represented as mean + SD. Statistical significance was examined by a one-way ANOVA test with Dennett's multiple comparison correction. *$P < 0.05$.

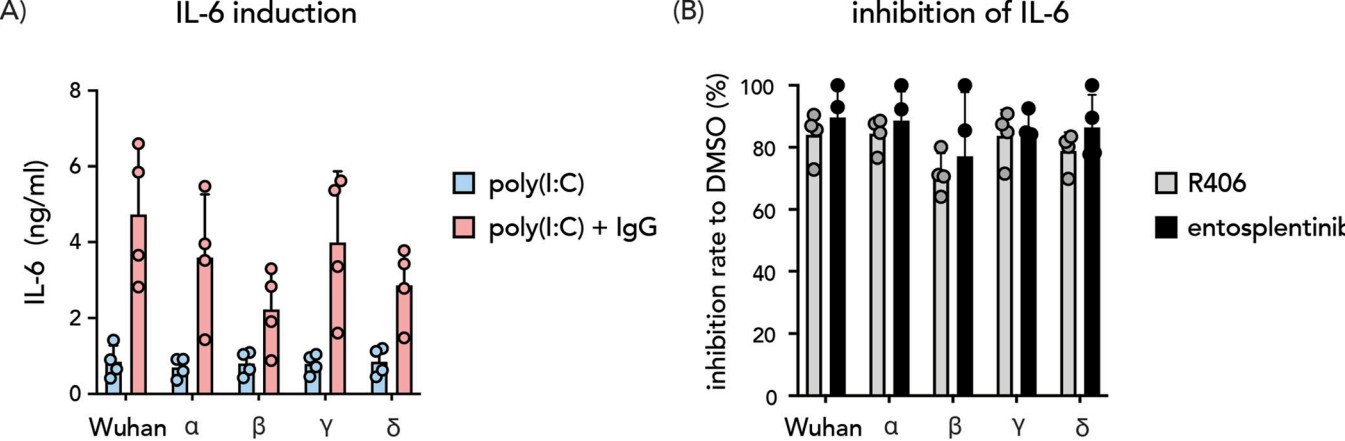

**Figure 6.  Antibody-induced inflammation by different SARS-CoV-2 variants of concern.**
**(A)** Immune complexes formed by spike proteins from variants of concern ($\alpha$, $\beta$, $\gamma$, $\delta$, and Wuhan strain) and a monoclonal antibody targeting a highly conserved epitope of the spike portion were used to simulate macrophages. IL-6 level was measured as the readout of the macrophage inflammatory response. **(B)** Inhibition rates of IL-6 production from macrophages treated with SYK inhibitors R406 and entospletinib compared with DMSO control (DMSO concentration 0.005%). Each dot represents cytokine production or inhibition rate by different macrophage donors (mean + SD).

spike-induced IL-6 was 45.6 nM with an efficacy of around 90% in a concentration of 1 $\mu$M. Hence, the steady-state serum concentration of entospletinib (C$_{trough}$ 3.02 $\mu$M to C$_{max}$ 6.54 $\mu$M) at a dose of 600 mg twice daily ([75]) would provide complete coverage of the IC$_{50}$ values throughout the 12-h dosing interval. In addition to the cytokine production inhibition, entospletinib can rescue the prolonged loss of HPMVEC barrier function and increased platelet adhesion mediated by anti-spike-induced macrophage hyperactivation. Endotheliopathy is associated with critical illness and death in COVID-19 ([76], [77]). Our findings are not only valuable for treatment targeting inflammation, but also have implications for strategies aimed at preserving endothelial function in COVID-19 and other related diseases. Furthermore, entospletinib counterbalances the hyperinflammation induced by anti-spike immune complexes across different SARS-CoV-2 VOCs. A recent study also showed that anti-spike IgG of SARS-CoV-1 could cause the antibody-dependent inflammation by alveolar macrophages, thereby deteriorating lung injury ([78]). As the mechanism of action of SYK inhibitors is through the inhibition of immune hyperactivation rather than through direct effects on coronaviruses, we are optimistic that entospletinib can be also applied for treatment of newly emerging variants and future coronaviruses.

Interestingly, in line with our previous findings ([33]), patient serum-derived immune complexes lead to substantially stronger induction of IL-6 compared with recombinant monoclonal IgG. IgG clonality, avidity, subclasses, and glycosylation patterns at the Fc domain all contribute to the activity of FcRs ([79], [80]). Our data indicate that dexamethasone is less potent in suppressing inflammation caused by serum-derived immune complexes, whereas SYK inhibitor R406 and entospletinib remain highly efficacious. It has been shown that the high titer and aberrant afucosylation of anti-spike IgG are two main serological characteristics in severe COVID-19 cases, which combined lead to hyperactivation of FcγRs ([34], [35], [37], [56]). Furthermore, under the prothrombotic environment in severe COVID-19 ([76], [81], [82]), aberrantly glycosylated anti-spike immune complexes can trigger platelet activation leading to thrombus formation. Ample evidence now supports the beneficial role of anti-platelet medication in COVID-19 treatments ([83], [84]). Therefore, as the altered glycosylation pattern of Fc tail on IgGs is transient in the early phase of seroconversion, the selective effect of entospletinib in counterbalancing thrombus formation against aberrantly glycosylated immune complex could be beneficial to prevent severe COVID-19. Yet, one major challenge with immuno-regulatory therapeutics against COVID-19 is the tailoring of treatments to the clinical course of the disease stages. SYK inhibition by fostamatinib has been shown to impair B cell development at the transitional stage but not mature B cell populations ([85], [86]). Because the proposed therapeutic effects of SYK inhibitors are dependent on spike-specific IgGs, appropriate timing for administrating these compounds is crucial.

It has been shown that immune complexes can also affect other cell types during COVID-19 disease progression. In severely ill patients, SARS-CoV-2 infection triggers soluble multimeric immune complex formation. These circulating immune complexes can activate monocytes via CD16 (FcγRIII) and promote immunopathology ([87]). Sera from severely ill COVID-19 patients contain high levels of immune complexes and activate neutrophil IL-8 production and CD11b expression via FcγRII (CD32) ([88]). Immune complexes also promote the degranulation of CD16+ T cells in severe COVID-19 ([89]). The activation of these highly cytotoxic CD16+ T cell population results in endothelial injury. Moreover, the CD16+ T cell proliferation and differentiation is driven by the cleaved complement product C3a ([89]), which is induced in macrophages upon immune complex stimulation ([90]). Evidently, anti-spike IgG with the aberrant glycosylation together with the predisposed pro-inflammatory milieu in the disease-prone patients could promote this uncontrolled vicious circle initiated by pulmonary macrophages. In light of these altered effector functions by immune complexes in various cell types in COVID-19, we propose that FcR-dependent activation is associated with disease severity on a systemic level instead of only in the (peri-)pulmonary region.

Even though IgG is the most prominent subtype present in the late stage of SARS-CoV-2 infection, other antibody isotypes such as IgA and IgM also shape the immune response after seroconversion. Particularly, IgA antibodies can promote the inflammatory response based on glycosylation pattern changes ([91], [92]). The detailed molecular mechanism of how different isotypes influence the SARS-CoV-2 specific immune response are currently still under investigation and out of the scope of this study. However, because

most activating FcRs in various immune cell types are dependent on downstream signaling through SYK, SYK inhibition could provide additional benefits against antibody-dependent inflammation beyond the antibody isotypes and cell types tested in this article ([93], [94]).

Although our data suggest SYK inhibitors are promising candidates for COVID-19 therapeutics, targeting other kinases in the FcγR signaling cascade did not yield similar results. PI3K is a group of signal transducer enzymes downstream of the FcγR-SYK pathway. Several studies have proposed the therapeutic potential of PI3K inhibitors in preventing uncontrolled inflammation and coagulation complications in COVID-19 patients ([95], [96]). However, our data indicate that PI3K inhibitors are less potent and efficacious than SYK inhibitors. The concentration required to reach 80% inhibition of anti-spike-dependent IL-6 by macrophages is high and can affect cell viability. Our observations of PI3K-induced effects on cell viability are in line with the already known problem of not fully studied early and late onset toxicity mechanisms of this class of drugs. In several clinical cases, the drug toxicity led to development of fatal adverse effects during treatment such as skin toxicity, autoimmune dysfunction, hypertension, and hyperglycemia ([97], [98]).

Furthermore, PI3K-γ/δ inhibitor duvelisib can induce macrophage repolarization toward a more pro-inflammatory phenotype in vivo ([99]). We also observed this pro-inflammatory activation by duvelisib in poly(I:C)-only conditions. Interestingly, in the presence of spike–IgG immune complexes, duvelisib suppresses IL-6 production by macrophages. As PI3K-δ-specific inhibitor idelalisib does not exert this differential regulation between TLR-dependent and anti-spike-dependent inflammation, the role of PI3K-γ is of great interest for further investigation.

In addition to the anti-inflammatory effects, blocking FcγR signaling in alveolar macrophages could halt disease progression through other mechanisms. Recent evidence shows that FcγRs mediate SARS-CoV-2 uptake by monocytes and tissue macrophages, which leads to pyroptosis and inflammasome activation that aborts virus proliferation, but aggravates systemic inflammation ([100], [101], [102]). As both SYK inhibitors fostamatinib and entospletinib are capable of blocking phagocytosis ([103], [104]), whether these compounds can curb SARS-CoV-2 uptake and subsequent pyroptosis in COVID-19 is of interest for further exploration.

A limitation of this study is that the used in vitro assay did not include IgG-opsonized cells that have been infected with live SARS-CoV-2. Yet, previous studies have indicated that IgG immune complexes that are generated with either plate-coated or cell-expressed spike proteins induce very similar inflammatory responses by human macrophages ([33]).

In conclusion, we show that small molecule SYK inhibitors specifically counteract the anti-spike-associated hyperinflammation while simultaneously preserving anti-viral immunity. We further demonstrate that entospletinib, the best candidate drug of this screening, may rescue anti-spike-induced endothelial barrier disruption and platelet adhesion. Moreover, we show that SYK inhibitors dampen inflammation triggered by different VOCs. Hence, entospletinib serves as a potential treatment option for halting COVID-19 progression independent of the virus variants. In conjunction with additional emerging evidence indicating the beneficial effect of

another SYK inhibitor fostamatinib, our work provides evidence for pursuing clinical trials to investigate repurposing entospletinib for counteracting COVID-19 pathology in severely ill patients.

# Materials and Methods

### Human subjects

Buffy coats were purchased from Sanquin Blood Supply in Amsterdam. All healthy donors provided written informed consent before blood donation. HPMVECs were collected from lung tissue obtained as waste material from a lobectomy performed at the Amsterdam UMC (location VU University Medical Center). Primary alveolar macrophages were obtained from broncho alveolar lavage

fluid as waste material from the ongoing DIVA study (Netherlands Trial Register: NL6318; AMC Medical Ethical Committee approval number: 2014_294). All volunteers of the DIVA study provided written consent form. The severe COVID-19 serum samples were collected by the Amsterdam UMC COVID19 Biobank according to approved protocols and in accordance with the Declaration of Helsinki.

### MDAMs

MDAMs were generated as previously described (33). In short, CD14$^+$ monocytes were isolated by Lymphoprep (Stemcell) isolation followed by CD14 magnetic beads purification via the MACS cell separation system (Miltenyi). The resulting monocytes were then differentiated with 50 ng/ml human M-CSF (Miltenyi) for 6 d in

**Key resources table.**

| Reagent or resource | Source | Identifier |
|---|---|---|
| Antibodies | | |
| COVA1-18 WT | P.J.M Brouwer et al (105) | doi:10.1126/science.abc5902 |
| COVA1-18 low fuc/high gal | Hoepel et al (33) | doi:10.1126/scitranslmed.abf8654 |
| COVA1-16 | P.J.M Brouwer et al (105) | doi:10.1126/science.abc5902 |
| Biological samples | | |
| Severe COVID-19 patient serum | Amsterdam UMC COVID19 Biobank | N/A |
| Primary alveolar macrophages | DIVA Study | NL6318 |
| Chemicals, peptides, and recombinant proteins | | |
| Human M-CSF | Miltenyi Biotec | Cat#130-096-491 |
| Recombinant human IL-10 protein | R&D Systems | Cat# 217-IL-025/CF |
| Recombinant SARS-CoV2-spike Wuhan Hu-1 protein | T.Caniels et al (60) | GenBank accession MN908947.3; doi:10.1126/sciadv.abj5365 |
| Recombinant SARS-CoV2-spike B.1.1.7 protein | T.Caniels et al (60) | doi:10.1126/sciadv.abj5365 |
| Recombinant SARS-CoV2-spike B.1.351 protein | T.Caniels et al (60) | doi:10.1126/sciadv.abj5365 |
| Recombinant SARS-CoV2-spike P.1 protein | T.Caniels et al (60) | doi:10.1126/sciadv.abj5365 |
| Recombinant SARS-CoV2-spike B.1.617.2 protein | M. van Gils et al (61) | doi:10.1371/journal.pmed.1003991 |
| Dexamethasone | Merck | Cat#D1756-25mg |
| Entospletinib (GS-9973) | Selleckchem.com | Cat# S7523 |
| R406 | Selleckchem.com | Cat#S1533 |
| Aleplisib (BYL719) | Selleckchem.com | Cat#S1815 |
| Idelalisib | MedChemExpres | Cat#HY-13026 |
| Duvelisib | MedChemExpres | Cat#HY-17044 |
| polyinosinic:polycytidylic acid (poly(I:C)) | Sigma-Aldrich | Cat#P1530 |
| Critical commercial assays | | |
| CD14 MicroBeads, human | Miltenyi Biotec | Cat#130-050-201 |
| ELISA MAX Standard Set Human IL-6 | BioLegend | Cat#430501 |
| Software and algorithms | | |
| GraphPad Prism version 9.4.0 | GraphPad Software | www.graphpad.com |
| R (v.4.1.3) | R Core Team (2022) | https://www.R-project.org/ |
| R package drc | Ritz et al (106) | doi:10.1371/journal.pone.0146021 |
| R package dr4pl | An et al (107) | doi:10.32614/RJ-2019-003 |

Iscove's Modified Dulbecco's Medium (Gibco) containing 5 % fetal calf serum (CAPRICORN) and gentamycin (Gibco). Total culture medium was refreshed after 3 d of culture. On day 6, M-CSF-differentiated macrophages were primed with 50 ng/ml IL-10 (R&D Systems) for 24 h. For further stimulation, cells were detached from the culture plates using TrypLE Select (Gibco).

## Coating

Stabilized recombinant SARS-CoV-2 spike protein and monoclonal antibodies (COVA1-16 and COVA1-18) were generated as previously described (60, 61, 105). To form immune complexes, 2 μg/ml spike protein diluted in PBS was incubated overnight on 96-well high-affinity plates (Nunc). To prevent unspecific binding, the plates were subsequently blocked with 10% FCS in PBS for 1 h at 37°C. After blocking, plates were incubated for 1 h at 37°C with diluted serum (2% in PBS) from severe COVID19 patients (Amsterdam UMC COVID19 Biobank) or 2 μg/ml monoclonal antibodies.

## Cell stimulation and inhibitor treatment

Selective small molecule inhibitors specifically against the SYK/PI3K signaling pathway were investigated (108). For repurposing purpose, only approved or investigational compounds in phase-III clinical trials were used in the screening assay. All inhibitors (dexamethasone [D1756; Merck], entospletinib [S7523; Selleckchem], R406 [S1533; Selleckchem], alpelisib [S2814; Selleckchem], idelalisib [HY-13026; MedChemExpress], duvelisib [HY-17044; MedChemExpress], were purchased in powdered form and dissolved according to the distributor's instructions). Macrophages were preincubated with inhibitors (or DMSO as a control) for 30 min at 37°C. After preincubation, macrophages were stimulated with 20 μg/ml poly-inosinic:polycytidylic acid (poly(I:C), Sigma-Aldrich) and seeded in a density of 50,000 cells/well in pre-coated 96-well plates in 200 μl/well medium.

## Enzyme-linked immunosorbent assay

To measure the IL-6 production, the supernatants of the stimulated cells were harvested after 24-h incubation. IL-6 concentration was determined using antibody pairs from U-CyTech Biosciences (Human IL-6 ELISA, CT744-20) or Biolegend (ELISA MAXTM Standard Set Human IL-6, 430501).

## Endothelial barrier function

Pulmonary microvascular endothelial cells (HPMVECs, passages 4 to 6) were seeded 1:1 in 0.1% gelatin-coated 96-well ibidi culture slides (96W10idf PET; Applied BioPhysics) for electrical cell-substrate impedance sensing, as previously described (109). In short, HPMVECs were maintained in culture in Endothelial Cell Medium (ScienCell) supplemented with 1% penicillin–streptomycin, 1% ECGS, 5% FCS, and 1% NEAA (Biowest). From seeding onward, electrical impedance was measured at 4,000 Hz every 5 min. HPMVECs were grown to confluence. After 72 h, ECM was removed and replaced by either complete ECM with DMSO or 1 μM entospletinib. After 2.5 h of pretreatment, the medium was removed and

replaced by the macrophage-conditioned media stimulated for 6 h as described above with poly(I:C) or in combination with patient serum. Three technical replicate measurements were performed for each condition. For every experiment, HPMVECs and macrophages obtained from different donors were used.

## Platelet adhesion on HPMVEC under flow

HPMVECs (passage 4 to 6) were seeded in 0.1% gelatin-coated six-channel μ-Slide VI 0.4 ibiTreat flow slides (#80606; ibidi) and cultured for 7 d. HPMVECs were preincubated for 2.5 h with complete ECM with DMSO or 1 μM entospletinib followed by 24-h treatment with macrophage-conditioned media as described above. On the day of perfusion, platelets were isolated from citrated blood from healthy volunteers, as previously described (110). Platelets were perfused for 5 min. After then, the phase-contrast and fluorescent images were taken using a 20× phase-contrast objective with an Etaluma LS720 microscope. Platelet adhesion was quantified in ImageJ (v. 1.53) by determining the platelet-covered area per field of view.

## In vitro thrombus formation

Blood samples were obtained from healthy donors that had given informed consent and using procedures approved by the University of Reading Research Ethics Committee and collected into vacutainers containing 3.8% (wt/vol) sodium citrate. Thrombus formation experiments were performed using microfluidic flow chips (Vena8; CellixLtd) coated with 5 μg/ml recombinant SARS-CoV-2 spike protein for 60 min at 37°C, washed, and then blocked with 10% FCS for 1 h at 37°C. The slides were then washed and treated with 10 μg/ml wildtype or lowly fucosylated and highly galactosylated COVA1-18 antibodies for 1 h at 37°C followed by 20 μg/ml vWF (Abcam) for 1 h. Thrombus formation was measured by perfusing citrated whole blood treated with 20 μg/ml vWF and either vehicle (DMSO) or entospletinib (1 μM) for 1 h through the flow chambers at 1000s-1 for 6 min before fixing with 10% formyl saline, staining with 2 μM DiOC$_6$, and then imaged by acquiring z-stacks using the 20× objective lens of a confocal Ti2 fluorescence microscope (Nikon).

## Calcein AM–propidium idodite (PI) cell membrane integrity assay

MDAMs were cultured in presence of the selected Syk or PI3K inhibitors for 24 h. After the inhibitor treatment, the medium was replaced with serum-free IMDM containing 1 μM Caclein-AM (56496-20X50UG; Sigma-Aldrich) and 3 μM PI reagent (P4170; Sigma-Aldrich). After 30 min incubation under culture conditions, membrane integrity and extracellular DNA content were determined by measuring fluorescence intensities at Ex/Em = 490/520 nm (Calcein AM) and Ex/Em = 530/620 nm (PI).

## Quantification and statistical analysis

Statistical significance of the data was performed in GraphPad Prism 9.4.0 (GraphPad). For $t$ tests comparing two sets of measurements, data were first examined with D'Agostino–Pearson normality test with $α = 0.05$ followed by paired or unpaired $t$ tests

according to the experiment design. The statistical examinations applied for each figure are stated in the legends. The half-maximal inhibitory concentration ($IC_{50}$) calculation was conducted in R (v.4.1.3) environment with R packages drc (106) and dr4pl (107).

## Data Availability

Further information and requests for data, resources, and reagents should be directed to and will be fulfilled by the corresponding author, Jeroen den Dunnen (j.dendunnen@amsterdamumc.nl). All data and codes reported in this article will be shared upon request.

## Supplementary Information

## Acknowledgements

We are grateful for the generous support from the Amsterdam UMC COVID-19 Biobank. J den Dunnen was supported by ZonMw (10430 01 201 0008), Amsterdam Infection and Immunity COVID-19 Grant (24184), AMC Fellowship (2015), European Union's Horizon 2020 Research and Innovation Programme (847551), and Innovative Medicines Initiative 2 Joint Undertaking Grant (831434). MPJ de Winther was supported by Amsterdam UMC, Amsterdam Cardiovascular Sciences, the Netherlands Heart Foundation (CVON GENIUS and GENIUSII 2017-20), Spark-Holding BV (2015B002 and 2019B016), Fondation Leducq (Transatlantic Network Grant 16CVD01), and ZonMW (09120011910025).

## Author Contributions

CE Geyer: data curation, formal analysis, validation, investigation, visualization, methodology, and writing—original draft.
H-J Chen: data curation, software, formal analysis, validation, investigation, visualization, methodology, and writing—original draft.
AP Bye: resources and investigation.
XD Manz: resources and investigation.
D Guerra: resources.
TG Caniels: resources.
TPL Bijl: resources.
GR Griffith: resources and investigation.
W Hoepel: resources and investigation.
SW de Taeye: resources.
J Veth: resources and investigation.
APJ Vlaar: resources.
G Vidarsson: resources and writing—review and editing.
H J Bogaard: resources and writing—review and editing.
J Aman: resources and investigation.
JM Gibbins: resources and writing—review and editing.
MJ van Gils: resources and writing—review and editing.
MPJ de Winther: conceptualization, supervision, funding acquisition, methodology, and writing—review and editing.
J den Dunnen: conceptualization, supervision, funding acquisition, methodology, and writing—review and editing.

## Conflict of Interest Statement

The authors declare that they have no conflict of interest.

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
