## [Reviewer comments · Life Science Alliance]

Life Science Alliance

Identification of new drugs to counteract anti-spike IgG-induced hyperinflammation in severe COVID19

Chiara Geyer, Hung-Jen Chen, Alexander Bye, Xue Manz, Denise Guerra, Tom Caniels, Tom Bijl, Guillermo Griffith, Willianne Hoepel, Steven de Taeye, Jennifer Veth, Alexander Vlaar, Gestur Vidarsson, Harm-Jan Bogaard, Jurjan Aman, Jonathan Gibbins, Marit van Gils, Menno de Winther, and Jeroen den Dunnen

DOI: <https://doi.org/10.26508/lsa.202302106>

Corresponding author(s): Jeroen den Dunnen, Amsterdam University Medical Centers and Menno de Winther, Amsterdam UMC

Review Timeline:

Submission Date:	2023-04-21
Editorial Decision:	2023-06-30
Revision Received:	2023-07-21
Editorial Decision:	2023-08-18
Revision Received:	2023-08-24
Accepted:	2023-08-29

Transaction Report:

June 30, 2023

Re: Life Science Alliance manuscript #LSA-2023-02106-T

Dr. Jeroen den Dunnen
Amsterdam UMC
Netherlands

Dear Dr. den Dunnen,

Thank you for submitting your manuscript entitled "Identification of new drugs to counteract anti-spike IgG-induced hyperinflammation in severe COVID19" to Life Science Alliance. The manuscript was assessed by an expert reviewer, whose comments are appended to this letter. We invite you to submit a revised manuscript addressing the Reviewer comments. In addition, please further contextualize these findings and mention that this identified inhibitor was not tested against live virus at this stage.

When submitting the revision, please include a letter addressing the reviewers comments point by point.

Thank you for this interesting contribution to Life Science Alliance. We are looking forward to receiving your revised manuscript.

Sincerely,

B. MANUSCRIPT ORGANIZATION AND FORMATTING:

Reviewer #1 (Comments to the Authors (Required)):

The authors have previously shown that high levels of anti-spike IgG with aberrant Fc tail glycosylation leads to hyperinflammation, greatly increasing the severity of COVID-19. The authors have identified a need for immune-regulatory therapeutics that counteract excessive inflammation while simultaneously minimizing inhibition of antiviral immunity. The authors developed an in vitro assay to screen for such drugs. Using this assay the authors identified entospletinib, an SYK inhibitor, as a promising drug candidate. They showed that entospletinib can counteract anti-spike-induced endothelial dysfunction and thrombus formation, markers of hyperinflammation. The authors also showed that entospletinib was effective against spikes from different SARS-CoV-2 variants.

Overall, this was a very well-written manuscript. The problem and approach were laid out very succinctly in the introduction. The figures in the results section were well-designed and easy to follow. The main points are well supported by the data and the results warrant the conclusions.

One minor comment I have would be to try and use the same scale for the graphs in Fig.1 and Fig. 2 whenever possible, to make comparisons easier for the reader.

Response to the reviewers' comments
LSA-2023-02106-T

We thank the reviewer for the feedback and comments and are glad to hear from the editor that they consider this work of interest for publication in Life Science Alliance.

Reviewer #1

The authors have previously shown that high levels of anti-spike IgG with aberrant Fc tail glycosylation leads to hyperinflammation, greatly increasing the severity of COVID-19. The authors have identified a need for immune-regulatory therapeutics that counteract excessive inflammation while simultaneously minimizing inhibition of antiviral immunity. The authors developed an in vitro assay to screen for such drugs. Using this assay the authors identified entospletinib, an SYK inhibitor, as a promising drug candidate. They showed that entospletinib can counteract anti-spike-induced endothelial dysfunction and thrombus formation, markers of hyperinflammation. The authors also showed that entospletinib was effective against spikes from different SARS-CoV-2 variants.

Overall, this was a very well-written manuscript. The problem and approach were laid out very succinctly in the introduction. The figures in the results section were well-designed and easy to follow. The main points are well supported by the data and the results warrant the conclusions.

One minor comment I have would be to try and use the same scale for the graphs in Fig.1 and Fig. 2 whenever possible, to make comparisons easier for the reader.

We thank the reviewer for this positive feedback and for the suggestion to change the scales. To allow an easier evaluation of the data despite the relatively large donor to donor differences we have included a scale showing the percentage of inhibition for each EC50 graph. Furthermore, we have adjusted the scale units for every scale in Figure 1 and 2 to a more uniform format showing all values in ng/ml. We hope these changes improved the design allowing a better comparison of the curves.

August 18, 2023

RE: Life Science Alliance Manuscript #LSA-2023-02106-TR

Dr. Jeroen den Dunnen
Amsterdam University Medical Centers
Center for Experimental and Molecular Medicine (CEMM)
Meibergdreef 9
Amsterdam 1105 AZ
Netherlands

Dear Dr. den Dunnen,

Thank you for submitting your revised manuscript entitled "Identification of new drugs to counteract anti-spike IgG-induced hyperinflammation in severe COVID19". We would be happy to publish your paper in Life Science Alliance pending final revisions necessary to meet our formatting guidelines.

- please add ORCID ID for corresponding (and secondary corresponding) author--you should have received instructions on how to do so
- please upload all figure files as individual ones, including the supplementary figure files
- all figure legends should only appear in the main manuscript file
- please add your main, supplementary figure, and table legends to the main manuscript text after the references section;
- please add a callout for Fig 5A, Fig 5B to your main manuscript text;
- please upload your Tables in editable .doc or excel format; -Tables should be numbered consecutively with Arabic numerals (1, 2, 3, 4); They can be included at the bottom of the main manuscript file or be sent as separate files.

A. FINAL FILES:

B. MANUSCRIPT ORGANIZATION AND FORMATTING:

Sincerely,

Reviewer #1 (Comments to the Authors (Required)):

The authors have adequately responded to my concerns. I recommend this manuscript for publication.

August 28, 2023

RE: Life Science Alliance Manuscript #LSA-2023-02106-TRR

Dr. Jeroen den Dunnen
Amsterdam University Medical Centers
Center for Experimental and Molecular Medicine (CEMM)
Meibergdreef 9
Amsterdam 1105 AZ
Netherlands

Dear Dr. den Dunnen,

Thank you for submitting your Research Article entitled "Identification of new drugs to counteract anti-spike IgG-induced hyperinflammation in severe COVID19". It is a pleasure to let you know that your manuscript is now accepted for publication in Life Science Alliance. Congratulations on this interesting work.

DISTRIBUTION OF MATERIALS:

Again, congratulations on a very nice paper. I hope you found the review process to be constructive and are pleased with how the manuscript was handled editorially. We look forward to future exciting submissions from your lab.

Sincerely,
